# A Cross-Sectional Study on the Acceptability of Implementing a Lung Cancer Screening Program in Belgium

**DOI:** 10.3390/cancers15010278

**Published:** 2022-12-31

**Authors:** Paloma Diab Garcia, Annemiek Snoeckx, Jan P. Van Meerbeeck, Guido Van Hal

**Affiliations:** 1Faculty of Medicine and Health Sciences, University of Antwerp, Universiteitsplein 1, 2610 Antwerp, Belgium; 2IQVIA RDS & Integrated Services Belgium NV/SA, Corporate Village-Davos Building, Da Vincilaan 7, 1930 Zaventem, Belgium; 3Department of Radiology, Antwerp University Hospital, 2610 Antwerp, Belgium; 4Department of Thoracic Oncology, Antwerp University Hospital, 2650 Edegem, Belgium; 5Laboratory of Experimental Medicine and Pediatrics, Infla-Med Center of Excellence, University of Antwerp, 2610 Antwerp, Belgium; 6Social Epidemiology and Health Policy, University of Antwerp, 2610 Antwerp, Belgium

**Keywords:** lung cancer screening, prevention, Belgium, acceptability

## Abstract

**Simple Summary:**

Lung cancer is the most common and deadliest cancer worldwide, and the number of new cases per year is expected to grow. Belgium is among the top 10 countries with most new cases of lung cancer in the world, with lung cancer incidence accounting for 11.8% of all cancers diagnosed and 23.8% of all cancer-related deaths. This study aimed to determine the overall acceptability of a lung cancer screening program in the Flemish population and the main factors that would influence the overall acceptability of such a program. Modeling of the results of a questionnaire distributed to the Flemish population showed 92% acceptability. Furthermore, policymakers should aim for it to be reimbursed, and campaigns should be gender-specific, focused on those with lower educational and socioeconomic status, along with investment in increasing total knowledge about lung cancer and about protective factors.

**Abstract:**

Lung cancer is the most common and deadliest cancer in the world, and its incidence is expected to grow. Nonetheless, this growth can be contained through smoking cessation programs and effective lung cancer screening programs. In 2018, Belgium had the seventh highest incidence of lung cancer in the world, with lung cancer incidence accounting for 11.8% of all cancers diagnosed and 23.8% of all cancer-related deaths that same year. The aims of this study were to determine the overall acceptability of a lung cancer screening program in the Flemish population and to determine the main factors that would influence the overall acceptability of such a program. A questionnaire-based cross-sectional study was performed in the Flemish population and distributed online and on paper. The results are presented with the variables of interest and the main outcome, i.e., the acceptability of participating in such a program if implemented. Odds ratios were used to compare acceptability between subgroups. A multivariate regression model was used to determine the key factors that would have the largest impact on the level of acceptability and, thus, on the possible efficiency of such a program. This study estimated that acceptability of participating in a lung cancer screening program was 92%. Irrespective of the smoking status, levels of acceptability were higher than 89%. The key factors which could significantly influence the acceptability of a lung cancer screening program were individuals with low education, low protective factor knowledge and total knowledge, and lung cancer screening reimbursement, which were significantly associated with acceptability (0.01, 0.001, 0.01, and 0.05 respectively). Low protective factor knowledge decreased the log odds of acceptability 3.08-fold. In conclusion, the acceptability of implementing a lung cancer screening program in Flanders seems to be extremely high and would be well received by all. When implementing such a program, policymakers should aim for it to be reimbursed, campaigns should be gender-specific, focused on those with lower educational and socioeconomic status, and there should be investment in increasing total knowledge about lung cancer and knowledge about protective factors.

## 1. Introduction

Lung cancer (LC) is globally one of the most common and deadliest cancers, and 75–80% of LC diagnoses are due to smoking [1]. LC has a 5 year survival of only 10–18% and represents 27% of all cancer-related deaths, making its death rate higher than the combined breast, colon, prostate, and pancreas cancer death rates [2,3,4]. Furthermore, 70% of diagnoses occur at a late stage (stage 4), leading to a low survival rate [5]. Late diagnosis is mainly due to the non-specificity of symptoms (cough, wheezing, hemoptysis, and dyspnea) and the asymptomatic nature of early-stage LC [1].

Every year, LC has a significant worldwide incidence of around 1.8 million new cases, with 1.6 million deaths as a result [6,7]. As of 2018, Belgium was classified as the country with the seventh highest LC incidence rate in the world [8]. At the country level, LC was the second most prevalent type of cancer in Belgium for men and women [8]. In 2018, LC accounted for most cancer-related deaths (23.8%) [8]. In Flanders, 10 times more men were diagnosed compared to Brussels and three times more men were diagnosed compared to Wallonia [9]. A similar trend could be seen for women [9].

Studies have suggested that LC will remain a major cause of death worldwide in the following decades, making smoking cessation and screening key [10,11]. The implementation of a high-quality screening program enables early diagnosis and, thus, a longer survival period and higher survival rate. 

The National Lung Screening Trial (NLST, USA) and the Nederlands–Leuvens Longkanker Screenings Onderzoek (NELSON, Belgium and Netherlands) are the main two randomized controlled trials that had enough power to prove lung cancer-specific survival when screening is used [12]. The NLST concluded that low-dose computed tomography (LDCT) allowed for early diagnosis of LC in high-risk individuals, producing a 20% reduction in LC specific mortality and a significant 6.7% reduction in all-cause mortality [5,13]. The NELSON trial, similarly, concluded that LDCT screening would result in a 26% reduction in cause-specific mortality. The NLST study led to the US implementing lung cancer screening (LCS), which is also considered in many European countries [14,15].

Although the beneficial evidence supporting LDCT as a screening tool for LC is high, it has only been applied to the population considered as high risk. To increase effectiveness, a positive benefit–harm ratio is needed, which depends on attracting the high-risk population and the number of individuals to be screened [5,16]. Many studies have shown the barriers research groups are faced with when trying to recruit participants [17,18]. 

One of the main challenges that LCS programs face is reaching their target group and ensuring that they make use of the program. Secondary challenges include lack of knowledge, unfamiliarity with the program, and practical issues such as distance [18]. Stigma surrounding LC is one of the major barriers that LCS programs have to face when recruiting smokers and ex-smokers [18]. A country-specific approach is essential for a successful and effective LCS program, as barriers may differ according to cultural differences.

As of 2020, the 2013 guidelines implemented by the United States Preventive Services Task Force have been updated regarding eligibility for LDCT screening: adults aged 50–80 years who have a 20 pack-year smoking history and currently smoke or have quit within the past 15 years [19,20]. In 2010, they observed a 3.3% participation rate, as well as low referrals by primary practitioners [17,19]. The significant lag in LCS uptake since it was implemented in 2010, initially due to stigma, pushed researchers in the USA to identify the cause [21]. These studies provided valuable insight into the implementation of such a program from the patient’s point of view, in order to increase uptake, the motivation to be screened or not, and the development of a more patient-centered screening process (improved shared decision making) [22]. 

Such studies are key to understanding the barriers to participation and preventing or minimizing them, thus enabling an effective campaign and maximizing program effectiveness [23,24].

In Belgium, a special task force was created to pave the way for LCS in Flanders. No in-depth studies have yet been conducted on the acceptability of the implementation of an LCS program. It would be beneficial to investigate barriers for participation before implementation of a Flemish LCS program. 

The aims of this study were to determine the overall acceptability of a LCS program in the Flemish population and to determine the main factors that influence the overall acceptability of such a program.

## 2. Materials and Methods

### 2.1. Ethics Approval

The study was approved by the ethics committee of the Antwerp University Hospital and the University of Antwerp, Belgium (Code 20/52/721). The study was performed in accordance with the Declaration of Helsinki (2013). Informed consent or e-consent (checkbox on online survey) was required from each participant before starting the survey. All answers were anonymous and securely stored in a password-protected server at the University of Antwerp.

### 2.2. Study Design and Setting

A questionnaire-based cross-sectional study was conducted. The online and paper questionnaires in Dutch were distributed from 27 January to 28 February 2021. The questionnaire was subdivided into four sections: background information, health information, beliefs and knowledge about lung cancer and screening, and screening program implementation. 

The study took place in Flanders, as the region is responsible for organizing its own cancer screenings. 

### 2.3. Data Collection

A convenience sample of the target population both eligible and noneligible for screening from the Flemish population was chosen according to specific criteria and informed about the “Enquête Longkankerscreening” survey (see Table 1). Never-smokers were included in the study sample as their viewpoint is key to determining the possible level of stigma and opposition, as well as how to remedy it. Throughout this study, acceptability refers to the acceptability of individuals to participate in a LCS program.

The online version allowed for a larger snowball effect (social media diffusion of QR code by participants) and, thus, considerably increased the number of respondents in a short time and made form completion more feasible. The overall completion time was 10–20 min. 

To achieve a large convenience sample in a short time, different distribution techniques were used for the dispersal of the online and paper versions of the questionnaire (see Table 2).

By having other organizations and clinics distribute the survey, we aimed to avoid bias and to reach a very wide range of participants from all socioeconomic backgrounds, with different genders and different smoking backgrounds. The option of the paper version allowed us to reach people with low socioeconomic background or minority groups which tend not to be included in LCS research.

### 2.4. Data Analysis

Firstly, data collected online from participants were checked for completeness, with the final output presented in an Excel sheet. Paper forms were checked for consent, and the information provided was transcribed to Excel with the online results and coded accordingly. Variables of interest for complete cases were selected to create a new dataset for analysis.

Secondly, the Excel software (Microsoft Office 365, 2013) was used to determine the overall level of acceptability of the population surveyed. A score per individual, for each knowledge table (risk factors, protective factors, and general knowledge) was then calculated, as well as a final total knowledge score. Knowledge scores were determined by inputting the number of correct answers each individual gave for each different knowledge table. Once scores were obtained, these were added per individual to obtain the final total knowledge score. Each knowledge table focuses on different topics as can be observed in Appendix A. For each type of knowledge score, a median was provided. We then stratified totals for each of the knowledge types into low, medium, and high based on the total number of questions (see Table 3 for stratification limits).A table with the descriptive summary statistics per percentage of our study sample was created, which included all variables studied and their subcategories (see Table 3).

Thirdly, R statistical software version 3.6.1 (R Core Team) was used for the main data analysis. A summary of the dataset was looked at to confirm dichotomization into acceptability and non-acceptability, as well as inclusion of all variables of interest (gender, education, income, family history of LC, asbestos exposure, involved in other screening, LCS reimbursement, inclusion in smoking cessation program (SCP), and participation in a SCP) and factorization into appropriate subgroups of categorical data (gender, income, education, smoking status, risk factor knowledge, protective factor knowledge, general knowledge, and total knowledge). 

A binary logistic regression was applied to determine if there was any significant (*p* < 0.05) correlation between acceptability and any of our variables. The intercepts were exponentiated to obtain the odds ratio of acceptability with a 95% confidence interval. 

The variables smoking status, education level, income, and gender underwent a two-way ANOVA test to determine whether any significant difference in acceptability existed between subcategories conforming to the variables. A chi-squared analysis was used to determine whether there was any multivariate correlation among all variables. 

Fourthly, a multiple logistic regression model was made to identify significant factors that determined overall acceptability of the implementation of a LCS program. Logistic regression allows for a binomial response variable and handles more than two explanatory variables simultaneously more efficiently, allowing to obtain odds ratios [25]. This kind of model allows us to not ignore the covariance among the variables and, thus, limit confounding effects and bias [26]. Model 1 was created with acceptability as the dependent variable. This model then underwent multiple logistic regression to obtain the initial Akaike information criterion (AIC) value. A stepwise procedure was then applied to identify the variables to create the full model and the final AIC value and determine whether any of the variables conforming to the full model significantly impacted the level of acceptability. The full model’s performance versus that of Model 1 was then tested through comparison of AIC, pseudo-r squared, deviance (*p* < 0.05), and likelihood ratio test (*p* < 0.05 for significance). Furthermore, the area under the curve of the full model was determined, and residuals were checked for overdispersion by plotting the ratio of deviance over degrees of freedom. As a last step, the mean probability of acceptability with the full model if implemented and the efficiency of the full model were determined.

## 3. Results

Following data transcription and transfer to Excel, we obtained a final sample of 511 individuals, of which 487 had no missing data. Having around 5% of participants with missing data did not significantly impact our results (see Appendix B, Figure A1). Out of the 511 individuals, 476 (94%) completed an online questionnaire and 35 (6%) completed a paper one.

### 3.1. Descriptive Statistics Results

The descriptive analysis was based on the 487 individuals forming our dataset. In Table 3, it can be seen that our sample was made of 36% males and 64% females. In terms of income, we observed a fairly similar distribution between the average-income (1501 to 3500 EUR/month) and high-income (>3501 EUR/month) groups (33% and 49%, respectively), although the low-income group (<1500 EUR/month) had quite a low representation (6%). This distribution between social classes was also expected due to the levels of education observed. Specifically, 76% of participants had a higher education level (at least an undergraduate degree), 20% had an average level of education (at least second-level secondary education), and 5% had a low level of education (maximum first-level secondary education). In terms of the proportionality of never-smokers and ex-/smokers it could be established that they were quite proportional (54% vs. 46%). General knowledge had a median of 7.00 out of 13, risk factor knowledge had a median of 6.00 out of 11, protective factor knowledge had a median of 3.0 out of 5, and total knowledge had a median of 16.00 out of 29 questions answered. Moreover, 87% of those surveyed were in favor of LCS being reimbursed by the health system, 62% had participated in other types of screening programs, 94% had not been exposed to asbestos, and 91% were in favor of including a SCP as part of the LCS program. The latter is significant since only 4% of individuals surveyed had ever participated in a SCP. In terms of assessing acceptability which was one of the major aims of this study, we observed that 91% were in favor of the implementation of a LCS program in Flanders. More specifically never-smokers showed 89.73% acceptability, while, for smokers, this was 96.72%, and, for ex-smokers, this was 92.04%. 

When comparing paper and online survey formatting, we can see significant differences in the following variables: gender, smoking status, and general knowledge. From those who filled out the survey online, 67% were female vs. 21% for the paper versions. In terms of smoking status, the online forms were filled out by 32% ex-smokers and 55% never-smokers, whereas the paper forms had 34% never-smokers and 59% ex-smokers, showing a clear inversing of percentages. Furthermore, 65% of those that filled out online forms had a medium level of general knowledge, whereas, on paper forms, 55% of individuals had a low level of general knowledge (see Appendix C, Table A2 and Table A3).

Table 4 provides us with the results in terms of odds ratio (95% confidence interval) of acceptability, for which we highlight the most important ones. The odds of males accepting the implementation of a LCS program was 0.86 times lower than that of females. The acceptability of an individual with an average income was 0.95 times lower than that of someone with a high income. A contrary observation can be seen in terms of education. The acceptability for someone with an average education level was 1.43 times higher than that for someone with a high education level. The acceptability for someone who has a LC history in the family was 1.47 higher than that for someone with no family history. Furthermore, the acceptability for someone who has been exposed to asbestos was 0.83 times lower than that of non-exposed participants. Compared to ex-smokers, the odds ratio for acceptability of smokers was 4.04 times higher. Individuals who underwent other types of screenings had an odds ratio of acceptability 1.53 times higher than that of non-screened participants. The level of acceptability for those with a medium score of general knowledge was equal to that of individuals with high scores. The acceptability for those with a low knowledge of risk factors was 1.04 times higher than that of individuals with a higher score. The acceptability for those with a medium knowledge score of protective factors was 0.50 times lower than that of individuals with a higher score. The acceptability for those with a low total knowledge score was 18.18 times higher than that of individuals with a higher total score. The acceptability when including LCS reimbursement was 2.73 times higher than when it was not included, while, for the inclusion of an SCP, it was 1.83 times higher, and, for previous participation, in an SCP it was 0.64 times lower. It should be further noted that the odds ratio of low protective factor knowledge and low total knowledge were significant compared to other factors included in this table. Odds were deemed significant since the 95% confidence interval included 1. Thus, it can be suggested that an association exists between these predictors and the level of acceptability. 

The two-way ANOVA test showed no significant differences between the subgroups in terms of *p*-value. Therefore, only one general model of acceptability was performed, which could be expected with the high overall acceptability obtained. Additionally, the chi-squared test showed that knowledge of protective factors was significantly correlated to acceptability with a *p*-value of 0.0002. 

### 3.2. Modeling Results

The binary multivariate logistic regression with acceptability as the binary outcome and all 14 variables resulted in Model 1 with an AIC of 297.21. Of the 14 variables, the protective factor knowledge and the total knowledge, specifically the low subcategories, were statistically significant with a *p*-value of 0.001 and 0.01, respectively.

The step AIC provided the full model with the variables that had the most impact on acceptability with an AIC of 277.57. Five variables were included in this model which are education, smoking status, protective factor knowledge, total knowledge, and LCS reimbursement. Of the five variables, low education, low knowledge of protective factors, and low total knowledge were statistically significant with *p*-values of 0.01, 0.001, and 0.01, respectively. Moreover, LCS reimbursement had a *p*-value of 0.05.

Once the full model and Model 1 were obtained, model performance checks were performed. The full model had a lower AIC than Model 1 (277.57 < 297.21). The full model had an AUC of 0.72 < 1. The pseudo-R squares were also compared between null and full models. The Nagelkerke test concluded that the full model demonstrated 12% more variability of the dataset, and the likelihood ratio test was significant with a *p*-value of 0.0007. When the full model was applied to the dataset, the mean probability of acceptability of the implementation of an LCS program in Flanders would be 92.13%; therefore, the full model had an efficiency of 91.79%.

## 4. Discussion

Overall acceptability of a possible Flemish LCS program was exceedingly high (92%). Smokers had an acceptability of 96.72%, compared to 92.04% of ex-smokers and 89.73% of never-smokers. Our data suggest that stigmatization by never-smokers might not be a problem in Flanders. Although smokers and ex-smokers are considered harder to reach, they represented 46% of our sample. This information might indicate that an LCS program could have a high uptake in our target group. Our findings are in accordance with Kellen et al.’s (2020) study in Belgium which concluded that an LCS program would be accepted by 83.6% of individuals and could achieve a high attendance rate (84.7% in current smokers and 82.2% in ex-smokers) [27]. The 10% differences seen in subgroups could be due to the selection criteria, study location, and formulation of the questions. A study performed in the United States (US) showed that individuals meeting the NELSON criteria had an overall acceptability of 77.3%, with 89% (7% lower than in our study) acceptability for current and 94% (2% higher than in our study) acceptability for former smokers [28]. Despite the cultural and inclusion criteria differences, the figures were almost overlapping between the US study and our own results.

In contrast to many LCS surveys, our female representation was high, accounting for 64% of our sample. Acceptability of females was shown to be higher, which suggests that campaigns for LCS participation should be more focused on the male population than the female one as odds of acceptability in males are 0.86 times lower. Further research should be performed if these results are to be replicated since many LCS surveys show that acceptability in males tends to be higher than in females.

Compared to other studies, this study included all categories of educational level and income, allowing the determination of odds ratios of acceptability for all three groups. Compared to high income, acceptability in average- and low-income groups was 0.95 and 0.45 lower, respectively. Thus, during planning and promotion of an LCS program in Flanders more focus should be given to the needs of individuals with a low income, as their odds of participation are lower and their income is a known risk factor for smoking and, thus, also for LC [29,30].

One of the unique characteristics of this survey is that it included knowledge scores on protective factors, risk factors, knowledge about LC, and total score. All scores were just barely over the average for each category. Campaigns should focus on increasing the knowledge of the Flemish population about risk factors specifically for each smoking status, as well as on protective factors. The latter is particularly relevant due to its dependent relationship with acceptability (*p* < 0.05). 

LCS reimbursement could be a key element of attraction for people to get screened, as it increased acceptability odds almost threefold and should be clearly stated in all the campaign information. 

Our optimized model identified the following key variables, which might influence Flemish acceptability: individuals with low and average education, individuals who are smokers and never-smokers, individuals with low and medium protective factor knowledge and total knowledge levels, and LCS reimbursement. Individuals with low education, individuals with low protective factor knowledge and total knowledge, and LCS reimbursement were significantly associated with acceptability (0.01, 0.001, 0.01, and 0.05, respectively). 

Inclusion of an SCP was not included in the final model as it had no effect on the improvement of acceptability. This is significant as our descriptive analysis showed that 91% of individuals are in favor of including this aspect in a LCS program; more importantly, when looking at only smokers, 75% were in favor of such an inclusion. These findings are supported by Kellen et al., (2020) who found that 71.8% of current smokers in Belgium would be willing to receive an SCP as part of the LCS program, which almost equates to our findings [27]. Since the proportions of acceptance of including SCP are quite similar, this is an indication that the results could be generalized to the whole of Belgium. 

One of the strengths of our survey is that the population sample was large and diverse. The use of different distribution techniques allowed a good recruitment of smokers and ex-smokers, which tend to have low participation in LC surveys. Online questionnaires might have made it easier for smokers to participate without feeling stigmatized. This survey was broad and touched upon a number of variables that could affect acceptability, providing a more in-depth analysis. These strengths increased our results and external validity, and they provided a more precise and detailed knowledge of factors important for LCS within the Flemish population at all levels. 

Some limitations can also be found. Our dataset had 24 omissions (<10%), characterized as missing at random. We attempted to limit selection bias by using different survey formats and distribution techniques in order to reach a wider range of the population and increase diversity. Nonetheless, representation of low-income individuals was still low (6%) and happy smokers were underrepresented. Moreover, the survey was based on a theoretical LCS program; therefore, effectiveness once truly implemented cannot be determined nor can the level of participation be guaranteed. Social desirability bias was limited by making the survey completely anonymous.

## 5. Conclusions

In conclusion, the acceptability of implementing a LCS program in Flanders was extremely high (92%) and would be well received by all. 

When implementing such a program, policymakers should guarantee reimbursement by the health system as it will play a key role in increasing acceptability and uptake. Campaigns should focus on those with lower education and income and be gender-specific. SCP should be recommended to smokers in order to avoid smokers from using screening as an excuse to continue to smoke. Additionally, the Flemish population should be further educated about protective factors, as increased knowledge significantly and dependably affects acceptability. In general, total knowledge about LC should be improved in Flanders as it barely surpasses the average, thus helping to reduce stigma in the general population and allowing more smokers and ex-smokers to be motivated to undergo screening. 

## Figures and Tables

**Table 1 cancers-15-00278-t001:** Study participant recruitment inclusion and exclusion criteria. This table contains the criteria on the basis of which participants are included or excluded. These criteria were determined following study design and aims.

Inclusion Criteria	Exclusion Criteria
Female and male	<50 years old of age
Resident in Flanders	Did not provide consent for participation
Smokers (eligible for screening)	Did not speak Dutch
Non/ex-smokers (eligible and noneligible for screening)	Diagnosed with lung cancer in the past and/or undergoing lung cancer treatment

**Table 2 cancers-15-00278-t002:** Detailed information on the distribution techniques used for the online and paper version of the survey.

Paper Distribution (Paper Version Survey)	Online Distribution (QR Code and Link to Survey)
In two pharmacies and Antwerp University Hospital	In two pharmacies and on their social media
	Webpages of the Association against Cancer and Tobacco Stop
	Newsletter of the ArtsenKrant sent to GPs and tobaccologists
	Posters projected on the monitors of Antwerp University Hospital and University of Antwerp campus
	University of Antwerp newsletter and social media
	Antwerp University Hospital webpage
	Facebook page of the Levensloopcomités (‘Lifecycle Committees’ in West Flanders)

**Table 3 cancers-15-00278-t003:** Descriptive statistics of variables without missing data collected and used for modelling (n = 487). This table contains the distribution of characteristics linked to the respondents of the survey. The relative proportion of respondents falling within a certain category is presented.

Variables	Proportion of Respondents	Median
*Gender*		
Male	36%	
Female	64%	
*Version*		
Online	94%	
Paper	6%	
*Income*		
Low (<1500 EUR/month)	6%	
Medium (1501–3500 EUR/month)	33%	
High (>3501 EUR/month)	49%	
*Level of Education*		
Low (maximum first-level secondary education)	5%	
Medium (at least second-level secondary education)	20%	
High (at least an undergraduate degree)	76%	
*Acceptability*		
Yes	91%	
No	9%	
*Lung cancer family history*		
Yes	24%	
No	76%	
*Asbestos Exposure*		
Yes	6%	
No	94%	
*Smoking Status*		
Never smoked	54%	
Smoker	13%	
Ex-smoker	33%	
*Involved in other screening*		
Yes	62%	
No	38%	
*In favour of Lung Cancer screening reimbursement*		
Yes	87%	
No	13%	
*Include Smoking cessation program*		
Yes	91%	
No	9%	
*Smoking Cessation Program participation*		
Yes	4%	
No	96%	
*General knowledge*		7.00
Low: [0; 6]	32%	
Medium: [6; 10]	63%	
High: [10; 13]	5%	
*Risk Factor Knowledge*		6.00
Low: [0; 5]	20%	
Medium: [5; 7]	52%	
High: [7; 11]	29%	
*Protective factors knowledge*		3.00
Low: [0; 2]	4%	
Medium: [2; 3]	89%	
High: [3; 5]	6%	
*Total Knowledge*		16.00
Low: [0; 10]	5%	
Medium: [10; 20]	84%	
High: [20; 29]	11%	

**Table 4 cancers-15-00278-t004:** Odds ratio of acceptability of implementing a lung cancer screening program in Flanders with a 95% confidence interval for all variables and their subcategories in the dataset with complete cases. This table shows the variables included in the study for odds of acceptability. OR signifies the odds ratio, while 2.50% and 97.50% refer to the limits conforming to the 95% confidence interval. The intercept is the odds of acceptability of implementing an LCS program independent of covariates. * Significant odds ratio.

	OR	Confidence Interval	
		2.50%	97.50%
Intercept	3.72	0.34	56.01
Gender male	0.86	0.4	1.89
Low income	0.45	0.11	2.29
Medium income	0.95	0.42	2.21
Not disclosed income	0.51	0.19	1.54
Low level of education	0.32	0.07	1.76
Medium level of education	1.43	0.54	4.37
Lung cancer family history	1.47	0.64	3.76
Asbestos exposure	0.83	0.23	4.17
Never-smoker	0.62	0.27	1.33
Smoker	4.04	0.74	48.88
Other screenings	1.53	0.73	3.18
Low general knowledge	0.88	0.13	5.10
Medium general knowledge	1.00	0.17	4.71
Low risk factor knowledge	1.04	0.32	3.43
Medium risk factor knowledge	0.87	0.33	2.13
Low protective factor knowledge *	0.05*	0.005	0.30
Medium protective factor knowledge	0.46	0.07	1.84
Low total knowledge *	18.18*	1.61	290.09
Medium total knowledge	2.73	0.68	10.54
Lung cancer screening reimbursement	2.23	0.84	5.42
Include smoking cessation program	1.84	0.54	5.26
Previous participation in a smoking cessation program	0.64	0.04	16.64

## Data Availability

Information is available from the corresponding author upon request.

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
