# Peer review of "A Cross-Sectional Study on the Acceptability of Implementing a Lung Cancer Screening Program in Belgium"

_cancers, 2022, doi:10.3390/cancers15010278_

Round 1

Reviewer 1 Report

The work attempted to provide insight into the acceptability of an hypothetical lung cancer screening program in the Flemish population.

My main concern about the paper the contestualization of the external validity of findings which I think should be better discuss in the paper. Is the population of respondents in some way representative of the target population of the hypothetical screening program? why subject non-eligible to the hypotetical screening program were also included in the survey? In case there is no evidence a representativeness I would suggest to conclude that a high acceptability was observed in the respondents of the survey, which are/are not/are partially representative of the target population of the hypothetical screening program.

It would be nice to also provide results stratified by respondents to paper-based vs on-line responder, at least in the supplementary material, in order to substanciate the statement reported in the method section (line 153-156) which I think should be better placed in the discussion section.

What about surveys with incomplete data? did you consider analyzing the available data from this subset and compare with the main study results? was missing data more frequent among on-line or paper based surveys?

Line 169 why you use mean?was the distribution normal?

Table 4 Please explain the meaning of "intercept" and how the reader should interpret it. The authors used a statistical significance test although it was barely mentioned the difference between significant and non-significant findings and what are the differences in terms of inerpretation of results.

Please add references for understanding the methodology used for the statistical analysis described at the end of the method section.

Line 204 Did you mean "5% of survey with missing data" instead? 

Line 295 I do not think the presented results can support the statement "which does not significantly impact our results".

Line 300 - 302 I would rather say that despite the cultural and inclusion criteria differences the figures were almost overlapping....

Was the knowledge score a custom-built tool specifically developed for this study? Or was it already used somewhere else?

Line 358 - 359 The autors concluded that there was an high acceptability in the Flanders, however this statement it is not supported by the presented data which analyse information from a specific subpopulation of Flanders which should be addressed/described in the conclusion.

Author Response

Dear Reviewer,

We would like to thank you for your remarks, which we all tried to take into account (see attachment). We are convinced that it certainly improved our manuscript. We also had our manuscript reviewed by a native speaker of English.

Reviewer 1:

  • Comment 1: My main concern about the paper the contestualization of the external validity of findings which I think should be better discuss in the paper. Is the population of respondents in some way representative of the target population of the hypothetical screening program? why subject non-eligible to the hypotetical screening program were also included in the survey? In case there is no evidence a representativeness I would suggest to conclude that a high acceptability was observed in the respondents of the survey, which are/are not/are partially representative of the target population of the hypothetical screening program.
  • Answer on Comment 1: The population of respondents is representative of the targeted population as 54% represented ex/smokers which will be the main population to be targeted as smoking is a major risk factor regarding lung cancer. The non-smokers opinion is quite important too as like evidence has shown a lot of stigma surrounds smoking and lung cancer furthermore such a screening program would be financed by public money thus having a good understanding of the general population opinion regarding a screening program for a disease whose main risk factor is smoking which is stigmatised is key. This would also let stakeholders know if such a program would be well received. As it will cause less issues for organisation but also to attract the target population which tends to be very hard to reach and would be more reachable if such a program will be well perceived and received by the public. Furthermore, such a program also targets non smokers who by other factors might be diagnosed with lung cancer. Including non-smokers is important as well to get a better understanding on the level of knowledge and to understand how much education will be needed to inform and help the public understand that lung cancer does not only affect ex/smokers.
  • Comment 2: It would be nice to also provide results stratified by respondents to paper-based vs on-line responder, at least in the supplementary material, in order to substanciate the statement reported in the method section (line 153-156) which I think should be better placed in the discussion section.
  • Answer on Comment 2: We did not stratify results based on online or paper response as only 6% of our respondents are paper respondents thus results would not truly be comparable. We have added the extra table in the supplementary section (see appendix C).

  • Comment 3: What about surveys with incomplete data? did you consider analyzing the available data from this subset and compare with the main study results? was missing data more frequent among on-line or paper based surveys.
  • Answer on Comment 3: 17 individuals who answered online had missing data (3.6%) and 7 individuals who answered via paper had missing data (20%). Thus, we found a higher proportion of missing data in paper respondents. We did do a missing data analysis (see Appendix B). We concluded that missing data had no significant impact.
  • Comment 4: Line 169 why you use mean?was the distribution normal?
  • Answer to Comment 4: Test of normality was not performed as data is categorical, which was confirmed as well when we performed the normality test on the knowledge score before categorization, median should have been stated instead and this has been corrected in table 3 and results/method section. Hence why a logistic regression was done since we have categorical variables.
  • Comment 5: Table 4 Please explain the meaning of "intercept" and how the reader should interpret it. The authors used a statistical significance test although it was barely mentioned the difference between significant and non-significant findings and what are the differences in terms of inerpretation of results.
  • Answer to Comment 5: See track changes in table 4 and lines 269-273
  • Comment 6: Please add references for understanding the methodology used for the statistical analysis described at the end of the method section.
  • Answer to Comment 6: See track changes in line 195 to 199 and references 25 and 26
  • Comment 7: Line 204 Did you mean "5% of survey with missing data" instead? 
  • Answer to Comment 7: corrected in Track Changes see line 212-214
  • Comment 8: Line 295 I do not think the presented results can support the statement "which does not significantly impact our results".
  • Answer to Comment 8: Missing data analysis which was performed can support the statement made in line 295. (See appendix B)
  • Comment 9: Line 300 - 302 I would rather say that despite the cultural and inclusion criteria differences the figures were almost overlapping....
  • Answer to Comment 9: see Track Changes
  •  
  • Comment 10: Was the knowledge score a custom-built tool specifically developed for this study? Or was it already used somewhere else?
  • Answer to Comment 10: The knowledge score was a custom built tool specifically developed for this study.
  • Comment 11: Line 358 - 359 The autors concluded that there was an high acceptability in the Flanders, however this statement it is not supported by the presented data which analyse information from a specific subpopulation of Flanders which should be addressed/described in the conclusion.
  • Answer to Comment 11: Antwerp is the biggest and most diverse city in Flanders. We had no limitations on the nationalities or backgrounds of respondents. We were able to obtain feedback almost equally from all backgrounds and genders. We do believe this study sample size was large enough to provide a good response and representation of the ideas of individuals living in Flanders. Furthermore the survey was shared through social media thus not limiting response to Antwerp. We do believe that the stament stands.

Kind regards,

Guido Van Hal

On behalf of all authors.

Reviewer 2 Report

-I don't totally understand the justification for including never-smokers in the survey. The authors write, "Non-smokers were included in the study sample as their viewpoint is key to determine possible level of stigma and opposition and how to remedy this." How is the level of stigma and opposition being determined in this case? Since the screening program will likely focus on high-risk populations, shouldn't we limit the analysis to those who are at high-risk? At the very least, the authors should consider examining correlates of acceptability separately for never smokers vs. current/ex-smokers (perhaps this is what the ANOVA test is doing, but it's not clear to me from the description). Also, the authors should be careful to differentiate between non-smokers and never smokers.  

-A bit more clarity around some of the measure descriptions would be helpful. Fore example, the authors write that, "A score per individual, for each knowledge table was then calculated as well as a final total knowledge score." How were these scores calculated and how were they categorized. I believe some of this information may be found in an appendix, but I think it should be included in the body of the manuscript as well. 

Author Response

Dear Reviewer,

We want to thank you for your comments, which we tried to take into account. We are convinced that it improved our manuscript.

Reviewer2:

- Comment1: -I don't totally understand the justification for including never-smokers in the survey. The authors write, "Non-smokers were included in the study sample as their viewpoint is key to determine possible level of stigma and opposition and how to remedy this." How is the level of stigma and opposition being determined in this case? Since the screening program will likely focus on high-risk populations, shouldn't we limit the analysis to those who are at high-risk? At the very least, the authors should consider examining correlates of acceptability separately for never smokers vs. current/ex-smokers (perhaps this is what the ANOVA test is doing, but it's not clear to me from the description). Also, the authors should be careful to differentiate between non-smokers and never smokers.

- Answer to Comment 1:  Justification to include non-smokers: see answer on comment 1 of Reviewer 1 above. The Anova test did indeed compare acceptability of such a study between all three smoking status groups. The anova test was used to compare levels of acceptability between sub-categories for the variables Smoking Status, Education level, Income, and Gender. Thus correlation with acceptability was compared between male and females; high, middle and low income; high middle and low levels of education; and between smokers, non-smokers and ex-smokers. Nonsmokers = never smokers >>adapted in the document so only never smokers is mentioned.

-Comment 2: A bit more clarity around some of the measure descriptions would be helpful. For example, the authors write that, "A score per individual, for each knowledge table was then calculated as well as a final total knowledge score." How were these scores calculated and how were they categorized. I believe some of this information may be found in an appendix, but I think it should be included in the body of the manuscript as well. 

-Answer to Comment 2: see Track Changes in the manuscript.

Kind regards,

Guido Van Hal

On behalf of all authors.
